# SARS-CoV-2 Positivity in Foreign-Born Adults: A Retrospective Study in Verona, Northeast Italy

**DOI:** 10.3390/life14060663

**Published:** 2024-05-22

**Authors:** Virginia Lotti, Gianluca Spiteri, Gulser Caliskan, Maria Grazia Lourdes Monaco, Davide Gibellini, Giuseppe Verlato, Stefano Porru

**Affiliations:** 1Microbiology Section, Department of Diagnostic and Public Health, University of Verona, 37134 Verona, Italy; davide.gibellini@univr.it; 2Occupational Medicine Unit, University Hospital of Verona, 37134 Verona, Italy; gianluca.spiteri@aovr.veneto.it (G.S.); mariagrazialourdes.monaco@univ.it (M.G.L.M.); stefano.porru@univr.it (S.P.); 3Unit of Epidemiology and Medical Statistics, Department of Diagnostics and Public Health, University of Verona, 37134 Verona, Italy; gulser.caliskan@univr.it (G.C.); giuseppe.verlato@univr.it (G.V.); 4Section of Occupational Medicine, Department of Diagnostics and Public Health, University of Verona, 37134 Verona, Italy

**Keywords:** foreign born, Italians, working adult population, SARS-CoV-2, COVID-19, nasopharyngeal swab

## Abstract

We compared SARS-CoV-2 positivity between the foreign-born adult working population and Italians living in the Verona area to investigate whether being a foreign-born adult could confer an increased risk of infection or lead to a diagnostic delay. The present study included 105,774 subjects, aged 18–65 years, tested for SARS-CoV-2 by nasopharyngeal swabs and analyzed at the University Hospital of Verona between January 2020 and September 2022. A logistic regression model was used, controlling for gender, age, time of sampling, and source of referral. A higher proportion of SARS-CoV-2 positivity in Italian (30.09%) than in foreign-born (25.61%) adults was reported, with a higher proportion of SARS-CoV-2 positivity in men than women in both cohorts analyzed. The difference in swab positivity among Italian and foreign-born adults was the highest in people aged 18–29 years (31.5% vs. 23.3%) and tended to disappear thereafter. Swab positivity became comparable between Italian and foreign-born adults during the vaccination campaign. Multivariable analysis confirmed the lower risk of swab positivity among foreign-born adults (OR = 0.85, 95% CI 0.82–0.89). In the Verona area, foreign-born adults showed a lower rate of SARS-CoV-2 positivity than the native population, likely because of underdiagnosis. Hence, public health should increase attention toward these particularly vulnerable populations.

## 1. Introduction

The severe acute respiratory syndrome coronavirus 2 (SARS-CoV-2) pandemic has increased the importance of evaluating epidemiological parameters to assess the effects of sickness on employees, including the occupational risk of infection, the availability and quality of occupational health services, the effectiveness and consistency of diagnosis, tracing, and follow-up procedures on cases and contacts at work, and the possibility to isolate and quarantine in case of need [1].

Italy was the first European country to be hit by the SARS-CoV-2 pandemic. As of January 2024, in Italy, almost 27 million inhabitants were SARS-CoV-2-positive at least once, and more than 195,000 died because of the infection (data reported on 7 February 2024) [2]. The prevalence and severity of COronaVIrus Disease 19 (COVID-19) were enhanced by the presence of chronic diseases, which were in turn associated with social determinants of health, including living and working conditions as well as access to qualified healthcare services. Indeed, COVID-19 has unevenly affected the poor, minorities, and a broad range of vulnerable populations, including foreign-born individuals, since it spreads easily in conditions of high-density populations, usually associated with less cultural inclination to medicalized care. Hence, the pandemic represented an important risk factor for the health of the migrant population in Italy, not only because of COVID-19 consequences but also because of the impact of pre-existing barriers in access to health services, leading to a slower diagnosis and therapy, which worsen the prognosis. Moreover, the COVID-19 pandemic has exacerbated some already existing disparities among weaker populations, including migrant populations, with an unequal impact of the pandemic on the working adult population, thus widening social gaps [3]. It is also well-established that employment status, working circumstances, and social security benefits are important health factors [4,5]. Interestingly, COVID-19 outbreaks in employment contexts were usually detected within health care, agriculture, services, and construction, where the migrant population is overrepresented [6], increasing the risk of infection for foreign-born adults and their families. In general, the migrant population was less tested than the general population, and in some cases, since access to the healthcare system is not always and constantly guaranteed for non-nationals or temporary workers, the risk of infection might have increased [7]. Moreover, according to the literature, the risk of SARS-CoV-2 infections among migrant populations compared with native groups is consistently higher, even though hospitalization rates and ICU admissions are reported to be lower [8]. On the other hand, foreign-born individuals are generally younger and healthier than the resident population (healthy migrant effect), so they should be less prone to severe COVID-19 [9].

### Aims

In the Veneto region, where the incidence and prevalence of SARS-CoV-2 infection have been particularly documented, the presence of the migrant population is very significant due to the high development of industries and economic activities that have attracted foreign workers for many years because of 3D (dangerous, dirty, and demanding/degrading) occupations. The main aims of our study were the following:-To analyze the incidence of SARS-CoV-2 infection among the migrant working adult population (18–65 years old) in comparison with Italians living in the Verona area;-To describe the pattern in access of foreign-born adults to COVID-19 health services and to assess whether being a migrant confers an increased risk of infection and leads to a diagnostic delay;-To give insights into the monitoring and intervention within public health by emphasizing the need to take more care of these more vulnerable populations in the case of epidemics;-To delineate priorities for research and intervention to enhance and improve occupational health surveillance of migrant workers.

## 2. Materials and Methods

### 2.1. Study Design and Sample Collection

A total of 105,774 subjects living in the Verona area, aged 18–65 years, tested for SARS-CoV-2 by nasopharyngeal swabs and analyzed at the Microbiology Laboratory at the University Hospital of Verona between 12 January 2020 and 15 September 2022, were considered. During the COVID-19 emergency, SARS-CoV-2 detection on nasopharyngeal swabs was foreseen because of provisions of national health authorities and conjugated at the local–regional level, such as screening and contact tracing programs offered. Moreover, for hospital admission, swabbing was also required; indeed, our sample also included swabs requested by the hospital department along with the hospital staff health surveillance samples. We stratified the studied population according to the fiscal code in Italians and foreign-born individuals; thus, we considered foreign-born subjects as coming from a foreign country and subjects born in Italy but without Italian citizenship. The sex–age distribution of Italians and foreign-born adults living in the Verona area in 2021 was obtained by the National Institute of Statistics (ISTAT) [10].

### 2.2. Inclusion Criteria

People 18 to 65 years of age (considered as the “working population”) who tested negative or positive for SARS-CoV-2 at least once by real-time PCR on nasopharyngeal swabs performed at the Microbiology unit of Azienda Ospedaliera Universitaria Integrata (AOUI) Verona from 12 January 2020 to 15 September 2022 were considered eligible to participate in this study.

### 2.3. Exclusion Criteria

Since we focused our analysis on the working population, subjects younger than 18 and older than 65 years (33% of all swabs performed) were excluded from this study as well as subjects with no valid fiscal code (0.8%) and those with undetermined results at the swab (1.4%).

### 2.4. Multiplex Real-Time RT–PCR for SARS-CoV-2 Detection

The analysis of nasopharyngeal samples for SARS-CoV-2 genome detection was performed by a commercial multiplex real-time RT–PCR platform, as described before [11].

### 2.5. Statistical Analysis

The significance of the association between the swab test and potential determinants was investigated by a chi-square test or Fisher’s exact test for categorical variables. The determinants considered were nationality (foreign-born nationals versus Italians), gender, age classes (18–29, 30–39, 40–49, 50–59, 60–65 years), time of sampling (before, during, or after the 1st vaccination campaign, which took place from 27 December 2020 to 31 July 2021), and source of referral (Verona Local Health Unit, Verona University Hospital, other). The proportion of swab positivity was computed with the corresponding 95% confidence interval, calculated by the Clopper–Pearson method.

Multivariable analysis was accomplished by a binary logistic regression model, where SARS-CoV-2 infection (negative or positive) was the response variable, nationality was the explanatory variable, and sex, age class, time of sampling, and source of referral were the potential confounders. The interaction between nationality and each of the potential confounders was also tested. The results were synthesized through odds ratios (ORs). The level of statistical significance was set at 5%, and confidence intervals (CIs) were calculated at 95%. Statistical analyses were performed using STATA software, release 17.0 (StataCorp, College Station, TX, USA).

## 3. Results

Of the 105,774 subjects analyzed, 19,398 (18.34%) were foreign-born adults. The Italians considered for the present study represented 17.5% (86,376/493,589) of the overall Italian population aged 18–65 years living in the Verona area, while the foreign-born adults represented 23.9% (19,398/81,266). The percentage of subjects included in this study remained rather stable irrespective of age among Italians, while it progressively increased from 18.2% among foreign-born adults aged 18–29 years to 35.4% among foreign-born adults aged 60–65 years (Figure 1). The sex and age distributions of the studied population are presented in Table 1. There were more women than men among both Italian and foreign-born adults, but the proportion of women was higher in the latter. The majority of foreign-born adults (55.6%) were in the middle age class (30–49 years), while most Italians belonged to either younger (19.1% aged 18–29 years) or older (42% aged 50–65 years) age classes. Approximately half of the swab testing was performed before the vaccination campaign; the proportion of tests performed during the first vaccination campaign was larger in foreign-born than in Italian adults. Most Italians were referred from the Verona Local Health Unit, while most foreign-born adults were referred from the Verona University Hospital itself.

A total of 30,959 (29.27%) swabs tested positive for SARS-CoV-2. The proportion of SARS-CoV-2 positivity was higher in Italian (30.09%, 95% CI 29.78–30.39%) than in foreign-born (25.61%, 95% CI 24.99–26.23%) adults. The proportion of SARS-CoV-2 positivity was significantly higher in men among both Italian and foreign-born adults (Italians: 32.23%, 95% CI 31.75–32.70%, *p* < 0.001; foreign-born: 29.05%, 95% CI 28.02–30.08%, *p* < 0.001) than in women (Italians: 28.39%, 95% CI 27.98–28.79%; foreign-born: 23.42%, 95% CI 22.65–24.19%) (Table 2). The proportion of SARS-CoV-2 positivity peaked between 40 and 59 years in both Italian and foreign-born adults, while the lowest proportion was recorded in the oldest age class among Italians and in the youngest among foreign-born adults.

Regarding the time of sampling, the proportion of SARS-CoV-2 positivity increased with time among Italians, while it peaked during the first vaccination campaign among foreign-born adults. SARS-CoV-2 positivity was nearly doubled among subjects from the Verona Local Health Unit with respect to those from Verona University Hospital (Table 2).

These results were substantially confirmed by the multivariable analysis. The odds of swab positivity were the highest among Italians, males, people aged 40–59 years, after the first vaccination campaign, and subjects referred from Verona University Hospital (Table 3).

The interactions between nationality and gender, age class, and time of sampling were all significant (*p* < 0.001). Regarding sex, the risk of SARS-CoV-2 positivity was higher in Italians than in foreign-born adults in both men (OR = 0.89, 95% CI 0.84–0.94) and women (OR = 0.83, 95% CI 0.79–0.87), but the difference was somewhat blunted among men (Figure 2A). The risk of swab positivity was significantly lower in foreign-born adults than in Italians in all age classes under 60 years, and the difference was the largest in people aged 18–29 years (OR = 0.77, 95% CI 0.70–0.84). On the other hand, no significant difference as a function of nationality was recorded in people aged 60–65 years (OR = 0.95, 95% CI 0.84–1.08; *p* = 0.457) (Figure 2B). Of note, a qualitative interaction was recorded between nationality and time of sampling. With respect to Italians, foreign-born adults were at lower risk of swab positivity both before (OR = 0.88, 95% CI 0.84–0.93) and after (OR = 0.67, 95% CI 0.62–0.72) the first vaccination campaign, while they presented a slight nonsignificant increase in risk (OR = 1.03, 95% CI 0.96–1.11) during the first vaccination campaign (Figure 2C). 

## 4. Discussion

This study analyzed the SARS-CoV-2 infection rate in Italian and migrant populations in Verona, one of the main cities of the Veneto region, which was one of the first areas where COVID-19 spread was detected. The wide observational period enabled us to evaluate the trend in SARS-CoV-2 positivity between Italian and foreign-born adults across the national vaccination program. In Italy, there are approximately 5 million foreign citizens residing (data as of 1 January 2023), representing up to 9% of the resident population, of whom 25% are concentrated in the northeastern regions. Veneto is the fourth Italian region by number of foreign-born residents, while it ranks in sixth position for the percentage of foreign-born residents in the total population. Indeed, foreign-born residents in Veneto on 1 January 2023 numbered 494,079 (10% of the total population), of whom 240,231 were male and 253,848 were female. In particular, the city of Verona counts 111,265 foreign-born residents (12% of the total population), 54,780 male and 56,485 female [12]. These figures are in line with the proportion of foreign-born adults tested in the present study: overall, the migrant population represents 12% of the total population analyzed.

Health and social–health service access for the migrant population is granted by the Italian National Health Service; however, cultural, language, and religious barriers remain, as well as bureaucratic difficulties, making access to health care difficult [13,14]. In Italy, migrant workers are primarily employed in manual jobs, often hired under precarious contracts with a higher risk of work injuries and occupational illnesses than natives [15] because of higher risk tolerance, thus determining a concentration of foreign-born adults in so-called 3D jobs [16,17,18,19,20]. Indeed, foreign-born adults were often engaged in occupational activities in which exposure to the virus was high because of precarious job sites and working conditions, with an increased risk of infection because of inadequate health and safety protection measures [21,22]. Moreover, they usually live in overcrowded housing conditions [23] in urban areas, which are known to be the most affected by the pandemic. Nevertheless, it was already reported that the impact of the COVID-19 pandemic on the migrant population in Italy differs from that observed in other European countries: in Italy, foreign-born individuals showed a lower incidence of infection than that observed in other European countries [23], possibly because of the lower access to swabs observed among foreign-born individuals [2]. Indeed, our results reported a higher proportion of SARS-CoV-2 positivity in Italians (30.09%) than in foreign-born adults (25.61%), in accordance with the current literature [24]. Recently, Ferroni et al. [25], based on an ad hoc analysis, reported a higher rate of swabs performed in Italians than in foreign-born individuals in Veneto, suggesting that the probability of identifying positive subjects in the Italian population vs. the migrant population could be affected. Again, limited access to basic health services and language and cultural barriers might explain this limited access to swabs for foreign-born individuals.

In our cohort of analysis, a different gender distribution between Italian and foreign-born adults was reported: women were represented more than men in both populations, with a higher proportion of women in the foreign-born population. These data are consistent with the census data (1 January 2023) stating a higher number of women than men in Italy, in Veneto and Verona, in total, and the migrant population [12]. We also reported a higher proportion of SARS-CoV-2 positivity in men both in Italians (32.23%) and foreign-born adults (29.05%) when compared with women (28.39% and 23.42%, respectively). These data are in accordance with the current literature reporting that men are more likely to be SARS-CoV-2 infected than women [25,26,27], and in cases of infection, they have a >50% higher risk of all-cause death, severe COVID-19 infection, and ICU admission than women [28,29]. Notably, differences in COVID-19 outcomes between male and female cases could be linked to biological, genetic, and behavioral sex-specific variations since sex is one of the variables that determine innate and adaptive immune responses [30]. 

In 2023, 24.1% of the Italian inhabitants were aged 65 years and older, 63.4% were aged between 15 and 64 years, and 12.5% were 14 years old or younger [12]. In our cohort, the majority of foreign-born adults (55.6%) were in the middle age class (30–49 years), while most Italians belonged to either younger (19.1%) or older (42%) age classes. According to the Organization for Economic Cooperation and Development (OECD), foreign-born individuals are younger on average than the native-born population [9]. This different age stratification of the migrant population compared with the native population may explain the differences reported in the cohorts we analyzed. Moreover, the higher capabilities of the middle-aged cohorts to break down both language and social barriers could be a possible explanation for the prevalence of middle-aged subjects found in the analyzed migrant population.

The 18–29 age class displayed, in Italians, a high SARS-CoV-2 swab positivity (31.46%), while in foreign-born adults, it represented the lowest (23.34%). The highest swab positivity was found for foreign-born adults in the 40–49 age class (27.15%) and for Italians in the 50–59 age class (31.50%). These results were influenced by the age distribution of our cohort, in which the majority of foreign-born adults belonged to the middle age class, while Italians belonged to either the younger or older age classes. Nevertheless, since the foreign-born population is known to be younger than the native-born population, it is highly likely that they may be less susceptible to developing serious health effects from COVID-19.

Moreover, according to our results, the age distribution of swab positivity differs between foreign-born and Italian adults in terms of risk of becoming ill from COVID-19 with respect to age groups and nationality, as denoted by the significant interaction between age and nationality (*p* < 0.001). Notably, the risk of swab positivity was significantly lower in migrants than in Italians in all age classes under 60 years, while no difference was detected in people aged 60–64 years. Moreover, a qualitative interaction between the time of sampling and nationality was noted. Foreign-born migrants had a marginally nonsignificant increase in risk (OR = 1.03, 95% CI 0.96–1.11) during the first vaccination campaign, but they were still at lower risk of swab positivity than Italians both before (OR = 0.88, 95% CI 0.84–0.93) and after (OR = 0.67, 95% CI 0.62–0.72). A possible explanation for these data might be that foreign-born individuals received their vaccinations later than Italians. In a recent study [25], the ratio of SARS-CoV-2 infection rates between Italians and foreign-born individuals was low, while that of hospitalization rates in cases of COVID-19 was markedly high, supporting a strong underdiagnosis of infection in migrant populations.

Some considerations invite caution in interpreting these results. Firstly, the sample under study, although representing about 20% of the general population in the Verona area, was not randomly drawn, so we cannot estimate incidence rates but rather the proportion of SARS-CoV-2 positivity. Second, “foreign-born” nationality is not tracked in the administrative flows of AOUI Verona. This prevents any opportunity to stratify the analyzed migrant population by specific geographical contexts of origin and, consequently, to disregard foreign-born epidemiological profiles, which vary depending on the country of origin. Thus, our study does not allow us to relate the risk detected with specific sources of the nationality of foreign-born adults. In addition, possession of a health insurance card by an applicant is another major limitation of this study, as the unassisted immigrant population, estimated at 8.7% [31], has not been considered. Moreover, we did not consider any information about the comorbidities of the subjects analyzed, which could alter the infection rate analysis. In the literature, a similar analysis of migrant residents in the Veneto region [25] reported differences in SARS-CoV-2 infection and hospitalization rates among migrant populations according to the geographic area of origin. Compared with our study, their limited period of analysis did not allow for examining the different waves of the pandemic. Moreover, they considered only SARS-CoV-2-positive subjects and not total swabs performed. Thus, our large sample size and the wide timeframe represent the main strengths of this study, which enabled us to stratify the positivity rate even according to the time of vaccination national program.

In summary, the evaluation of the impact of the COVID-19 pandemic on the migrant population of the Verona area showed a lower rate of swab positivity in comparison with the native population, although the former is exposed to occupational and sociodemographic factors that might increase the risks. Thus, a likely underdiagnosis of SARS-CoV-2 infection in migrant populations is present, leading to the need for public health to increase attention toward these particularly vulnerable populations by enhancing vaccination policies, contact tracing, and testing. Furthermore, since SARS-CoV-2 significantly impacted foreign-born workers, making them more vulnerable, it is imperative that health and safety practices be implemented in every workplace through risk assessment, health surveillance, and fitness for work targeted at foreign-born workers. This will enhance their working conditions and promote their health while adhering to corporate social responsibility principles. In future studies, the analysis carried out in this study could be enriched by adding some clinical data, such as comorbidities, vaccination status, and geographical areas of origin, thus considering further cultural and social characteristics. This would enable us to obtain more robust data and to appraise whether the differences that emerged in the present study are confirmed in broader contexts.

## Figures and Tables

**Figure 1 life-14-00663-f001:**
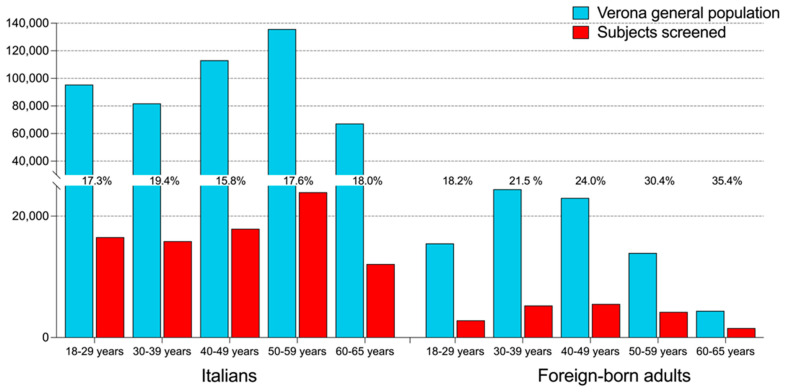
Age distribution of Italian and migrant subjects screened versus the Verona general population. Red columns indicate the age distribution of Italians and foreign-born adults considered for the present study, and blue columns indicate that of the whole population living in the same area. The percentages of subjects included in the present study over the entire population of the corresponding age class are also shown.

**Figure 2 life-14-00663-f002:**
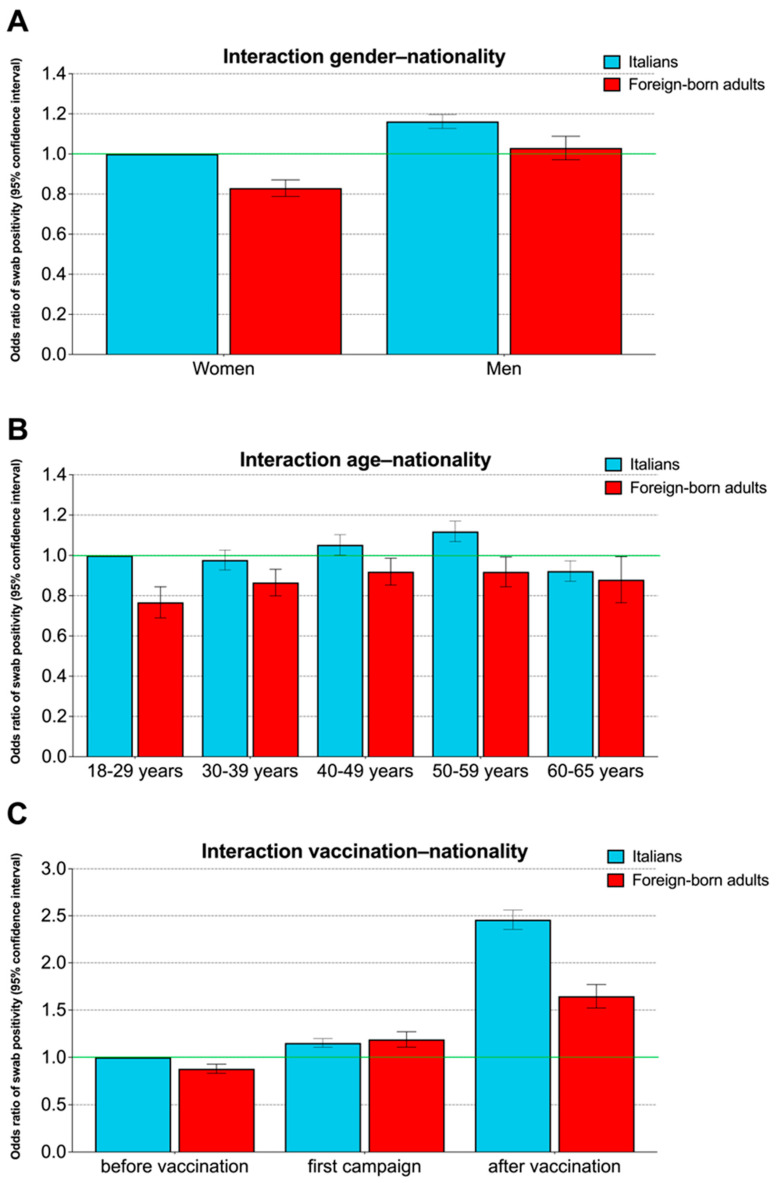
Interaction effects on swab positivity of nationality and either gender (**A**), age class (**B**), or time of sampling (**C**). The odds ratios and the corresponding 95% confidence interval were computed by a multivariable logistic model.

**Table 1 life-14-00663-t001:** Gender and age distribution of the Italian and migrant population that underwent swab testing from January 2020 to September 2022. Counts are reported with percent frequencies in parentheses. Significant differences are highlighted in bold.

	Total *n* = 105,774	Italians *n* = 86,376	Foreign-Born *n* = 19,398	*p*-Value
Gender (Female)	59,896 (56.63%)	48,062 (55.64%)	11,834 (61.01%)	*p* < 0.001
Age (years) (median, p25–p75)	45 (33–55)	46 (33–56)	42 (34–51)	*p* < 0.001
Age class (years)				*p* < 0.001
18–29	19,351 (18.29%)	16,528 (19.13%)	2823 (14.55%)	
30–39	21,148 (19.99%)	15,885 (18.39%)	5263 (27.13%)	
40–49	23,446 (22.17%)	17,918 (20.74%)	5528 (28.50%)	
50–59	28,163 (26.63%)	23,937 (27.71%)	4226 (21.79%)	
60–65	13,666 (12.92%)	12,108 (14.02%)	1558 (8.03%)	
Time of sampling				*p* < 0.001
Before vaccination	52,552 (49.7%)	43,395 (50.2%)	9157 (47.2%)	
During the first campaign	24,679 (23.3%)	19,585 (22.7%)	5094 (26.3%)	
After the first campaign	28,543 (27.0%)	23,396 (27.1%)	5147 (26.5%)	
Source of referral				*p* < 0.001
Verona Local Health Unit (ULSS 9)	53,847 (50.9%)	44,896 (52.0%)	8951 (46.1%)	
Verona University Hospital (AOUI)	50,687 (47.9%)	40,472 (46.9%)	10,215 (52.7%)	
Other	1240 (1.2%)	1008 (1.2%)	232 (1.2%)	

The significance of differences was assessed by Fisher’s exact test or the chi-squared test for categorical variables and by the Wilcoxon–Mann–Whitney rank sum test for the quantitative variable (age).

**Table 2 life-14-00663-t002:** Influence of gender and age class on swab positivity in the Italian and migrant populations.

	Italians *n* = 86,376	Foreign Born *n* = 19,398
Swab (Positive)	25,991 (30.09%)	4968 (25.61%)
Gender		
Male	12,347 (32.23%)	2197 (29.05%)
Female	13,644 (28.39%)	2771 (23.42%)
	*p* < 0.001	*p* < 0.001
Age class (years)		
18–29	5199 (31.46%)	659 (23.34%)
30–39	4529 (28.51%)	1270 (24.13%)
40–49	5538 (30.91%)	1501 (27.15%)
50–59	7541 (31.50%)	1145 (27.09%)
60–65	3184 (26.30%)	393 (25.22%)
	*p* < 0.001	*p* < 0.001
Time of sampling		
Before vaccination	12,204 (28.1%)	2265 (24.7%)
During the first vaccination campaign	6014 (30.7%)	1496 (29.4%)
After the first vaccination campaign	7733 (33.2%)	1207 (23.5%)
	*p* < 0.001	*p* < 0.001
Source of referral		
Verona Local Health Unit (ULSS 9)	17,422 (38.8%)	3315 (37.0%)
Verona University Hospital (AOUI)	8486 (21.0%)	1630 (16.0%)
Other	83 (8.2%)	23 (9.9%)
	*p* < 0.001	*p* < 0.001

**Table 3 life-14-00663-t003:** Determinants of SARS-CoV-2 positivity investigated by a logistic regression model.

	Odds Ratio (95% CI)	*p*-Value
Nationality		
Italian	1 (Reference)	
Foreign-born	0.85 (0.82–0.89)	*p* < 0.001
Gender		
Female	1 (Reference)	
Male	1.18 (1.14–1.21)	*p* < 0.001
Age class (years)		
18–29	1 (Reference)	
30–39	1.00 (0.96–1.04)	0.973
40–49	1.07 (1.03–1.12)	0.001
50–59	1.13 (1.08–1.17)	*p* < 0.001
60–65	0.95 (0.90–1.00)	0.034
Time of sampling		
Before vaccination	1 (Reference)	
During the first vaccination campaign	1.19 (1.15–1.23)	*p* < 0.001
After the first vaccination campaign	2.35 (2.26–2.44)	*p* < 0.001
Source of referral		
Verona Local Health Unit (ULSS 9)	1 (Reference)	
Verona University Hospital (AOUI)	0.16 (0.13–0.19)	*p* < 0.001
Other	0.28 (0.27–0.29)	*p* < 0.001

## Data Availability

The datasets generated during the current study are not publicly available, because they contain sensitive data to be treated under data protection laws and regulations. Appropriate forms of data sharing can be arranged after a reasonable request to the last author.

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
