# Peer review of "SARS-CoV-2 Positivity in Foreign-Born Adults: A Retrospective Study in Verona, Northeast Italy"

_life, 2024, doi:10.3390/life14060663_

Round 1

Reviewer 1 Report

Comments and Suggestions for Authors

Dear Authors,

Your research is very interesting. However I have several comments.

By “foreign-born” you mean both those who have an Italian citizenship and those who do not have it?

Why did you exclude 65+ population from the analysis?

Using the age (a numerical variable) as a categorical (i.e. recoding it into age groups) always decrease the power of the statistical tests. I would suggest using the age as a number.

“Binary logistic regression” is more correct than just logistic regression.

The age would be more useful if it is included as a numerical variable in the binary logistic models.

Figure 1: You could add a right axis to show foreign-born in order to increase the height of the columns and make them clearer

One again why the age should not be recoded: Pearson chi-square overestimate the presence of a relationship between the variables when the sample is so large. I would suggest to compare the mean or median age of both groups (and add appropriate measure of dispersion as well)

Page 5, lines 193-194: please add the p-value as well. The entire paragraph needs clarification about the differences: were they significant or not (lines 194-199).

Table 2 and 3: the entire analysis would benefit if you use the age as a number.

The discussion is clear and written very well. A symbol appeared at page 10, line 327 (‘t’).

Please see the highlighted text in the pdf.

Author Response

Dear Reviewer,

We sincerely thank You for your suggestions. We have revised the text based on your comments, deepening the impact of the findings. Please find attached a point-by-point response.

Sincerely yours,

Virginia Lotti

Reviewer 2 Report

Comments and Suggestions for Authors

The manuscript is impressive and addresses a key issue of inequity in health care implementation. 

The document is well-written in general, although several instances of phraseology will require adjustment. 

In lines 326-328, "Moreover, according to our results, the age distribution of swab positivity differs between foreign-born and Italians t, as denoted by the significant interaction between age and nationality (p<0.001)." needs to be rewritten to fill in missing wording.

Also, lines 343-349 require a number of fixes as denoted by [ ]: "This prevents [any opportunity] to stratify the analyzed migrant population by specific geographical contexts of origin and consequently to disregard foreign-born epidemiological profiles, which vary depending on the country of origin. Thus, our study does not allow [us] to relate the risk detected with specific sources of nationality of [the] foreign-born. In addition, to possess[ion of] the health insurance card by the applicant turns out to be another major limitation of this study, as the unassisted immigrant population, estimated at 8.7% (31), has not been considered."

The major substantive deficiency of the manuscript is that it provides little, if any, meaningful discussion of implications of the findings for policy and practice. Such implications, which are essential for the article to have broader impact, are likely to be a fairly simple extensions of what has been summarized.

Comments on the Quality of English Language

As noted in the comments a number of wording and syntax issues need to be addressed, but in general the manuscript is in good shape regarding English.

Author Response

(The authors gave the same response as above.)
